# C²-AFCL: Cross-task Calibration for Asynchronous Federated Continual Learning

## Abstract

Federated Continual Learning (FCL) aims to empower distributed devices to learn a sequence of tasks over time. However, existing FCL research largely relies on the impractical assumption of synchronous new task arrival. This overlooks the reality of asynchronous user behavior and system latencies, forcing more efficient clients to endure costly inactivity. The practical necessity of an asynchronous method gives rise to Asynchronous Federated Continual Learning (AFCL). The server constantly receives a mixture of updates from clients at different time steps, leading to a catastrophic task drift that corrupts the global model and prevents effective learning. In this paper, we introduce a novel Cross-task Calibration framework called **C²-AFCL** that is the first to tackle task drift at a semantic level within an Asynchronous FCL setting. Its core is a two-stage orthogonal calibration mechanism. First, intra-client calibration uses task-aware caches to mitigate variance from local client drift. Second, and more critically, inter-task interference calibration dynamically estimates an interference subspace from historical task knowledge. New updates are orthogonally projected to isolate and remove components that conflict with this subspace, preserving previous knowledge while learning new tasks. Extensive experiments show that C²-AFCL significantly outperforms existing methods, demonstrating robust and efficient learning in dynamic federated environments.

## 1 Introduction

Federated Learning (FL) enables collaborative model training on massive fleets of edge devices while preserving user privacy McMahan et al. (2017); Wang et al. (2023a); Liu et al. (2024). In recent years, this privacy-centric approach has attracted considerable research interest, leading to successful applications in areas such as recommendation systems Yang et al. (2020); Li et al. (2024d) and intelligent healthcare Xu et al. (2021); Nguyen et al. (2022a).

However, many existing works on FL assume a static environment where training data remains fixed. In real-world applications, these devices must adapt to evolving environments, a scenario addressed by Federated Continual Learning (FCL), where a sequence of tasks is learned over time. This dynamic process frequently results in a sharp decline in performance on previously learned tasks Yang et al. (2024); Wang et al. (2024a), a well-known issue termed catastrophic forgetting Ganin et al. (2016). In addition, due to the federated context environment, different tasks may further introduce data drift, which in turn affects the convergence of the global model.

To address these challenges, Federated Continual Learning (FCL) has emerged as a feasible solution by enabling clients to learn from evolving data streams without forgetting previous knowledge Wang et al. (2024c); Li et al. (2025b). Various strategies have been explored to this end Chen et al. (2025); Liang et al. (2024). Some approaches focus on data replay, either by training a generative model to reconstruct samples from previous tasks Qi et al. (2023); Wuerkaixi et al. (2023) or by selectively caching important exemplars for rehearsal Li et al. (2024a). Other methods tackle specific scenarios like class-incremental learning by introducing specialized loss functions to handle class imbalances Dong et al. (2022; 2023a). Knowledge distillation has also been employed, using supplementary data on both the server and clients to transfer knowledge and preserve model performance over time Ma et al. (2022a). The authors in Li et al. (2025c) aim to save computation costs and enhance data privacy by improving synaptic intelligence algorithms without sample replay.

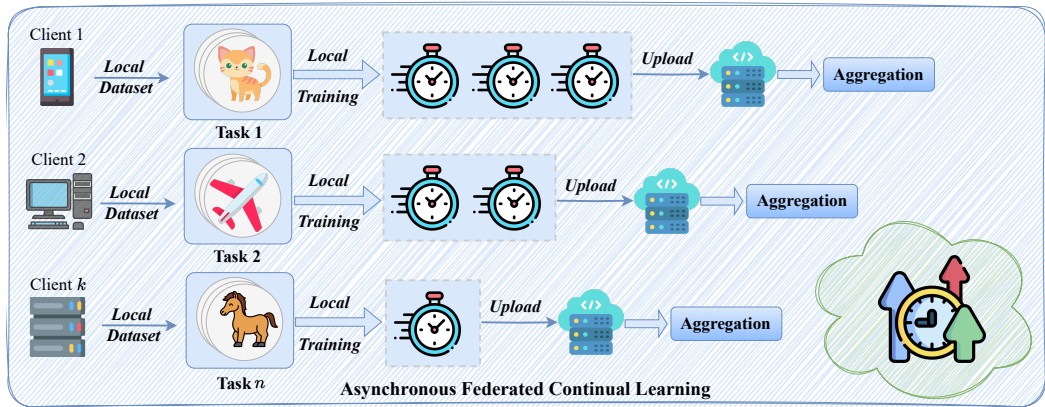

Figure 1: The illustration of asynchronous federated continual learning (AFCL). AFCL allows clients to handle different tasks at different times without waiting for unified aggregation, reducing waiting and improving efficiency.

Although these methods are all effective in handling catastrophic forgetting, a critical and often overlooked flaw in existing FCL research is the unrealistic assumption of synchronous task arrival, that all clients encounter and begin a new task at the same moment Li et al. (2025a). In practice, task arrival is inherently asynchronous. Different clients will start new tasks at different time steps due to varied user behavior, network latency in receiving task data, or device availability. Forcing synchronization in such a setting would compel faster clients, who are ready for the next task, to remain idle and wait for the slowest clients to catch up. This introduces substantial downtime and negates the core efficiency benefits of FCL. Therefore, an asynchronous operational method is not merely an option but a practical necessity for realistic FCL.

Recently, AFCL has attracted some attention, but not extensively. Shenaj et al. (2023) was the first to focus on this scenario by using fractal pre-training, a contrastive prototype-based loss for clients, and a modified server aggregation strategy to address catastrophic forgetting with a workshop research. Li et al. (2025a) surveyed hundreds of existing works and pointed out this research direction as a potential future work. However, these studies have not yet systematically explored AFCL, mainly due to the unprecedented challenge of *task heterogeneity*. At any given time, the server may receive a mixture of updates from clients at different time steps. This temporal misalignment of tasks, compounded by Non-IID data heterogeneity, induces a dual drift:

- **Client Drift:** The conventional inconsistency between local and global models caused by Non-IID data and update staleness.
- **Task Drift:** A more severe issue where the global model collapses between conflicting tasks from different clients, ultimately failing to learn any single task effectively and leading to catastrophic forgetting.

To tackle this challenge, we propose a novel cross-task calibrated AFCL framework **$C^2$-AFCL**. Our core idea is to elevate update calibration from a statistical level to a semantic level. We no longer treat each streaming task in isolation but actively manage and leverage the relations between them. The key is an innovative two-stage orthogonal calibration and aggregation process. We first maintain a task-aware update for each client. By computing the difference between a new update and the client's last contribution for the same task, we first eliminate the local variance arising from client's specific data and system delays. Then, the server leverages the historical task updates of all clients to dynamically construct an interference subspace that represents knowledge from previous tasks. Before any new update is aggregated, it is orthogonally projected onto this subspace. Its projection is identified and isolated, while its orthogonal component is used to update the model. In this manner, $C^2$-AFCL can explicitly decouple new knowledge from potential conflicts at the fine-grained update level, thereby striking a superior balance between plasticity and stability.

Through extensive experiments on two types of incremental learning tasks (Class-Incremental / Domain-Incremental) with four datasets and various settings, we verify both the effectiveness and

robustness of $C^2$-AFCL, and outperform the state-of-the-art baselines by up to xx.xx% in terms of average test accuracy. Our main contributions are as follows:

- We pioneer in motivating and defining the problem of the asynchronous federated continual learning as the practical paradigm for FCL. Moreover, we identify the task heterogeneity as its core challenge and provide a deep in-depth analysis.

- We propose $C^2$-AFCL, a novel framework that resolves the AFCL problem through a two-stage orthogonal calibration, elevating update calibration from statistical variance reduction to semantic knowledge management.

- We theoretically analyze the convergence for $C^2$-AFCL, which proves the same convergence rate as traditional FL baselines.

- Extensive experiments on multiple FCL benchmarks demonstrate that our method significantly outperforms existing techniques on key metrics.

## 2 BACKGROUND AND RELATED WORK

**Asynchronous Federated Learning.** Federated Learning (FL) is a machine learning paradigm allowing multiple clients to collaboratively develop a shared global model by training on their respective local datasets while maintaining data privacy Li et al. (2020); Wang et al. (2023b); Li & Wang (2019). A foundational algorithm in this domain is FedAvg McMahan et al. (2017), which refines a central model through the periodic aggregation of parameters from locally trained models. A significant hurdle for conventional FL methods like FedAvg is the unrealistic assumption that clients will upload the updates and do the global aggregation at the same time. To overcome the efficiency bottlenecks of synchronous FL, FedAsync Xie et al. (2019) introduced a weighted averaging scheme that adapts the contribution of an update based on its degree of staleness. FedBuff Nguyen et al. (2022b) employs a buffering strategy, where the server waits to collect a sufficient number of updates before aggregation to smooth the training process. More recently, $CA^2FL$ Wang et al. (2024b) proposed a cache-based calibration method where the server caches each client's latest update and uses it to calibrate newly received updates, effectively reducing the variance caused by data heterogeneity. However, these works focus on asynchronous updates under the same task, and this paper investigates asynchronous updates across different tasks, which directly and significantly affect the performance of model aggregation and are therefore more challenging.

**Continual Learning.** Continual Learning (CL), also known as incremental learning, equips a model with the ability to sequentially acquire knowledge from a continuous stream of tasks without catastrophically forgetting previously learned information Hsu et al. (2018); van de Ven & Tolias (2019). This learning paradigm encompasses various settings, including task-incremental learning Dantam et al. (2016); Maltoni & Lomonaco (2018), class-incremental learning Rebuffi et al. (2017); Yu et al. (2020), and domain-incremental learning Mirza et al. (2022); Churamani et al. (2021). Methodologies in CL are generally grouped into three families: replay-based Rebuffi et al. (2017); Liu et al. (2020), regularization-based Jung et al. (2020); Yin et al. (2020), and parameter isolation methods Long et al. (2015); Fernando et al. (2017). In this paper, we explore the intersection of federated and continual learning, where each client trains the local model with streaming new tasks.

**Federated Continual Learning.** Federated Continual Learning (FCL) addresses the challenge of learning from sequential tasks within a federated setting, focusing on enabling the global model to adapt to new information while preserving previous knowledge. As an emerging field, FCL has seen pioneering efforts such as the work by Yoon et al. (2021), which tackles Task-IL by using separate masks for each task, thereby requiring task labels at inference time for personalization. Another line of research Bakman et al. (2023) prevents parameter overwriting by projecting updates for different tasks onto orthogonal subspaces. Other approaches employ knowledge distillation with a surrogate dataset at both the server and client levels to transfer knowledge Ma et al. (2022b). More recently, an importance-aware sampling method was proposed in Li et al. (2024b;a), which selectively stores samples based on their contribution to local and global distributions to mitigate catastrophic forgetting. While some studies have expanded FCL to applications beyond image classification Jiang et al. (2021); Dong et al. (2023b), others have explored dynamic network architectures where clients train multiple personalized models, isolating or merging them based on task similarity Li et al. (2024c). However, these works are based on a fundamental assumption that all clients receive new tasks and

start training at the same time, which limits the practical deployment of such methods. In this work, we relax this assumption and investigate asynchronous federated continual learning.

## 3 PROBLEM FORMULATION AND PRELIMINARIES

We consider a federated continual learning setting composed of a central server and a set of $K$ clients. The system is designed to learn from a sequence of data tasks, denoted as $D = \{D^1, D^2, \ldots, D^t\}$, where $D^t$ denotes the data available at time steps $t$. The training for each task can encompass multiple rounds of communication between the server and clients. For any given task $t$, each client $k$ possesses a local dataset $D_k^t = \{(x_i, y_i)\}$, where $x_i$ is a data sample and $y_i$ is its corresponding label from the cumulative label space $Y^t$. Thus, the local data $D_k^t$ can feature samples from classes seen in prior tasks $\{Y^1, Y^2, \ldots, Y^{t-1}\}$ as well as new classes introduced at stage $t$. Let $w$ represent the classification model. The global model at stage $t$ is denoted by $w^t$, while $w_k^t$ is the local model for client $k$. The central objective is to train a unified global model $w^t$ that performs well across the entire sequence of tasks, effectively capturing the data distribution from all learned stages. This goal can be formally stated as:

$$\min_{w^t} \sum_{j=1}^{t} \sum_{k=1}^{K} \frac{1}{|D_k^j|} \mathbb{E}_{x \sim P(x|y) \in D^j} \mathcal{L}(f(x; w_k^j), y), \quad \text{where} \quad w^t = \sum_{k=1}^{K} p_k^t w_k^t. \tag{1}$$

where $\mathcal{L}$ denotes the cross-entropy loss, and $p_k^t$ is the aggregation weight. The formulation above assumes a *synchronous* update protocol, where the server must wait for a designated group of clients to return their local models before aggregating them.

Based on this, we then formulate the framework for an *asynchronous* protocol. In this paradigm, the server updates the global model as soon as it receives an update from any single client, rather than waiting for a full cohort.

Let $\tau$ index the version of the global model at the server, denoted by $w^\tau$. When client $k$ is ready for training, it downloads the current global model, say version $w_k^\tau$. After completing its local training on task data $D_k^t$, it sends its updated parameters $w_k^t$ back to the server. By the time this upload arrives, the server's global model may have already been updated by other clients and advanced to version $\tau$, where $\tau \geq t$. The server immediately integrates the received model:

$$w^{\tau+1} = (1 - \eta)w^\tau + \eta w_k^t. \tag{2}$$

where $\eta \in (0, 1]$ is a mixing parameter that functions as a server-side learning rate, controlling the influence of the incoming client update. The degree of staleness for this update is given by the difference $\tau - t$. On the client side, the local objective must be adapted to handle both the current task and the preservation of previous knowledge. Upon receiving $w_k^\tau$, client $k$ optimizes its local model by minimizing a composite loss function over its data $D_k^t$:

$$\min_{w_k^t} \mathbb{E}_{(x,y) \in D_k^t} [\mathcal{L}(f(x; w_k^t), y) + \lambda \Omega(w_k^t, w_k^\tau)]. \tag{3}$$

where the term $\Omega$ is a continual learning regularizer, weighted by a hyperparameter $\lambda$, which penalizes deviations from the downloaded model $w_k^\tau$ on parameters critical for previous tasks, thereby combating catastrophic forgetting.

## 4 C$^2$-AFCL: MITIGATING DRIFT VIA DUAL ORTHOGONAL CALIBRATION

To address the task heterogeneity challenges in AFCL, we propose the C$^2$-AFCL method. The key idea is to decompose any incoming update into a component beneficial for learning the new task and a component potentially harmful to previous tasks. This is achieved through a two-stage orthogonal calibration process before aggregation, which first mitigates client drift and then resolves task drift. The first stage targets client drift by normalizing updates against historical updates, thereby reducing variance from statistical heterogeneity. Then, we tackle task drift by projecting these calibrated updates onto a dynamically estimated subspace of previous task knowledge. We illustrate the workflow in Algorithm 1.

---

**Algorithm 1:** C$^2$-AFCL

---

**Input** : $R$: communication rounds; $K$: number of clients; $\eta$: learning rate; $\{T^t\}_{t=1}^n$: distributed dataset with $n$ tasks; $w^t$: global model parameters for the $t$-th task.
**Output:** $\{w_1^t, w_2^t, \ldots, w_K^t\}$: personalized target models for each client.

1 **for** $r = 1$ **to** $R$ **do**
2    Server randomly selects a subset of clients $S_t$ and sends $w^t$ to them.
3    **for** *each selected client* $k \in S_t$ **in parallel do**
4      Client $k$ downloads $w^t$ and trains on its task $T_k^t$, obtaining raw update $\mathbf{u}_k^t = w_k^t - w^t$.
5      Send the raw update $\mathbf{u}_k^t$ to the server.
6    **end**
     // Intra-client calibration (mitigate client drift)
7    Retrieve the last cached update $u_k^t$ for the same task;
8    Compute calibrated update: $\delta_k^t = \mathbf{u}_k^t - u_k^t$;
9    Update cache $u_k^t \leftarrow \mathbf{u}_k^t$.
     // Inter-task orthogonal calibration (mitigate task drift)
10    Server maintains the interference subspace $\mathcal{F}_t = \mathrm{Col}([a_1, a_2, \ldots, a_{t-1}])$, where $a_j$ are average updates of past tasks;
11    Compute projection matrix $\mathcal{P}_t = b_d b_d^\top$ from top-$d$ SVD basis;
12    Decompose calibrated update: $\delta_{\mathrm{pre},k}^t = (\mathbb{I} - \mathcal{P}_t)\delta_k^t$, $\delta_{\mathrm{int},k}^t = \mathcal{P}_t \delta_k^t$;
13    Keep only the preserving component $\delta_{k,t}^{\mathrm{pre}}$;
14    Server aggregates safe updates: $w_{r+1}^t = w_r^t + \frac{\eta}{|S_t|} \sum_{k \in S_t} \delta_{k,t}^{\mathrm{pre}}$.
15 **end**

---

## 4.1 Intra-Client Calibration for Client Drift Mitigation

The first stage of our framework is designed to tackle client drift, a phenomenon where a client's local update deviates significantly from the direction beneficial to the global model. In the AFCL scenario, this drift depends on two main factors: (1) Statistical heterogeneity, where each client's unique Non-IID data distribution pulls its local model towards a local optimum, and (2) System heterogeneity, where asynchronous updates computed on global models introduce temporal misalignment and error. A raw update from a client is therefore a noisy and biased signal. Simply averaging these raw updates would introduce significant variance into the global model, hindering convergence and stability.

In this paper, we first shift the paradigm from aggregating absolute updates to aggregating relative updates. This helps to distinguish the contributions of different tasks and prevents severe knowledge conflicts. Specifically, we cache a task-aware update $U_k = \{u_k^1, u_k^2, \ldots, u_k^t\}$ on the server for each $k$ and $u_k^t$ stores the last raw model update for the $t$-th task. When the server receives a new update $\mathbf{u}_k^t$, it retrieves the corresponding historical update $u_k^t$ from the cache. The intra-client calibrated update $\delta_k^t$ is then computed as their difference: $\delta_k^t = \mathbf{u}_k^t - u_k^t$. This operation isolates the client's learning progress during its most recent training round. By subtracting the previous update, we aim to cancel out the slowly-varying, static components of the client's update vector that are attributable to its fixed data distribution and consistent system characteristics. The resulting $\delta_k^t$ is a more accurate representation of the client's immediate learning trajectory, effectively reducing the variance caused by both statistical and systemic heterogeneity. After calibration, the cache is updated with the new update and prepares it for the next cycle.

## 4.2 Inter-Task Orthogonal Calibration for Task Drift Mitigation

While intra-client calibration mitigates the statistical noise of the update, it does not address its potential to conflict with semantic knowledge from previous tasks. The core technical motivation for our second stage is to prevent an update for a new task $T_k^t$ from destructively interfering with the consolidated knowledge of previous tasks $\{T_k^j | j < t\}$. Such interference occurs when the update $\delta_k^t$ contains components parallel to the critical gradient directions of previous tasks.

To resolve this, we propose a mechanism to identify and neutralize these harmful components through orthogonal projection. We construct an online interference subspace $\mathcal{F}_t$, which represents the accumulated knowledge of previous tasks. The server maintains a global average task update, $\mathcal{A} = \{a_1, a_2, \ldots, a_t\}$, where each vector $a_t = \frac{1}{N} \sum_{i=1}^{N} a_t^i$ serves as a proxy for the $N$ update direction for the previous $t$-th task. Then, the interference subspace $\mathcal{F}$ is defined as the column space of the matrix whose columns are the historical global vectors for all previous tasks:

$$\mathcal{F}_t = \text{Col}([a_1, a_2, \ldots, a_{t-1}]), \ \ \mathcal{B}_d = [b_1, b_2, \ldots, b_d] \in \mathbb{R}^{D \times d}. \tag{4}$$

For numerical stability and to focus on the most significant interference directions, we compute an orthonormal basis $\mathcal{B}_d$ for this subspace via Singular Value Decomposition (SVD), where it contains the $d$ principal left-singular vectors. The corresponding projection matrix onto this subspace is given by $\mathcal{P}_t = b_d b_d^\top$. Using this projection matrix, we decompose the calibrated update $\delta_k^t$ into an interference component, which lies within the subspace and is potentially harmful to previous tasks; and a preserving component, which is orthogonal to the subspace and represents novel knowledge for the $t$-th task:

$$\delta_{\text{int},k}^t = \mathcal{P}_t \cdot \delta_k^t, \ \ \delta_{\text{pre},k}^t = \delta_k^t - \delta_{\text{int},k}^t = (\mathbb{I} - \mathcal{P}_t)\delta_k^t. \tag{5}$$

For the final global aggregation, the server only aggregates the preserving components from clients working on the same task. These safe updates are collected in a task-specific buffer and used to update the global model:

$$w_{r+1}^t = w_r^t + \frac{\eta}{K} \sum_{k=1}^{K} \delta_{\text{pre},k}^t. \tag{6}$$

where $r$ denotes the communication round, this process ensures that the global model is updated only with information that is minimally disruptive to previously learned tasks.

## 5 THEORETICAL ANALYSIS

We first state the standard assumptions underpinning our analysis and then present a rigorous theoretical foundation for the algorithm's convergence.

**Assumption 5.1** ($L$-Smoothness.) For all tasks $t \in \{1, 2, \ldots, T\}$, the global loss function $F^t$ and each client's local loss function $F_k^t$ are $L$-smoothness. For any model parameters $x, y \in \mathbb{R}^D$, there exists a constant $L > 0$ such that (This also applies to the global loss function $F^t$):

$$\|\nabla F_k^t(x) - \nabla F_k^t(y)\| \le L\|x - y\|.$$

**Assumption 5.2** (Bounded Variance.) The clients' stochastic gradients are unbiased and have bounded variance. For any client $k$ and task $t$, its stochastic gradient $g_k^t(x)$ is an unbiased estimator of the true gradient $\mathbb{E}[g_k^t(x)] = \nabla F_k^t(x)$, and its variance is bounded by $\sigma^2$; and the variance of local gradients across clients is bounded by $\sigma_g^2$:

$$\mathbb{E}\|g_k^t(x) - \nabla F_k^t(x)\|^2 \le \sigma^2, \ \ \frac{1}{K} \sum_{k=1}^{K} \|\nabla F_k^t(x) - \nabla F^t(x)\| \le \sigma_g^2.$$

**Assumption 5.3** (Bounded Update Norm.) The expected norm of the raw updates generated by clients after local training is bounded. There exists a constant $G > 0$ such that $\mathbb{E}\|\delta_k^t\|^2 \le G^2$. This ensures that the norm of the cached vectors $u_k^t$ is also bounded.

These three assumptions are the most fundamental ones in the theoretical analysis of federated optimization, making the theoretical analysis of FL possible Li et al. (2019); Zhao et al. (2018).

**Assumption 5.4** (Bounded Asynchronous Delay.) The delay of client updates is bounded. There exists a constant $\tau_{max}$ such that for any update submitted by a client at any round, its delay $\tau$ satisfies $0 \le \tau \le \tau_{max}$, and we bound the maximum delay.

**Lemma 5.5** (Projection Drift Bound Yu et al. (2015).) Let $\mathcal{A}_t^{(s)}$ denote the covariance matrix used to construct the interference subspace at SVD update time $s$ for the $t$-th task, and define the increment $\mathcal{E}_t^{(s)} := \mathcal{A}_t^{(s+1)} - \mathcal{A}_t^{(s)}$ and the projection drift accumulation $\Delta_{\text{proj}}^t := \sum_{s=1}^{S-1} \|\mathcal{P}_t^{(s+1)} - \mathcal{P}_t^{(s)}\|_F^2$.

Assume that for every $s$ the $d$-th and $(d + 1)$-th eigenvalues of $\mathcal{A}$ are separated by a positive eigen gap $\delta_t^{(s)} := \lambda_d(A_t^{(s)}) - \lambda_{d+1}(A_t^{(s)})$, and denote $\delta_{\min}^t := \min_s \delta_t^{(s)} > 0$. Then the following hold:

$$\|\mathcal{P}_t^{(s+1)} - \mathcal{P}_t^{(s)}\|_F^2 \leq \frac{8d\,\|\mathcal{E}_t^{(s)}\|_2^2}{(\delta_t^{(s)})^2}, \text{ hence } \Delta_{\text{proj}}^t \leq \sum_{s=1}^{S-1} \frac{8d\,\|\mathcal{E}_t^{(s)}\|_2^2}{(\delta_t^{(s)})^2} \leq \frac{8d}{(\delta_{\min}^t)^2} \sum_{s=1}^{S-1} \|\mathcal{E}_t^{(s)}\|_2^2. \quad (7)$$

**Theorem 5.6** (Convergence Analysis.) Suppose Assumptions 5.1-5.4 hold, let the learning rate $\eta \leq 1/(4L)$ and $F^t(w^\star) = \arg\min_w F^t(w)$. Then after $R$ asynchronous aggregations with total $K$ clients, the global model $w_r^t$ of our method will satisfy:

$$\frac{1}{R} \sum_{r=0}^{R-1} \mathbb{E}\big\|(\mathbb{I} - \mathcal{P}_t)\nabla F^t(w_r)\big\|^2 \leq \frac{4\big(F^t(w_0) - F^t(w^\star)\big)}{\eta R} + 6L^2\eta^2\tau_{\max}^2 G^2 + 6\sigma_g^2$$

$$+ \frac{2L\eta\sigma^2(3L\eta\tau_{max}^2 + 1)}{K} + 3G^2\Delta_{\text{proj}}^t. \quad (8)$$

Consequently, if $\eta = O(1/\sqrt{R})$, the first three terms vanish at the standard $O(1/\sqrt{R})$ rate. The additional projection drift term is $O(1/T)$ if the projection is updated once every $T$ iterations ($S \approx R/T$). Thus, the algorithm can converge to an expected projected first-order stationary point. We provide the detailed proof for both the lemma and the theorem in Appendix B.

# 6 EXPERIMENTS

## 6.1 EXPERIMENT SETUP

**Datasets.** We evaluate our method under two federated incremental learning settings with heterogeneous data partitions, leveraging four benchmark datasets: (1) Class-Incremental Learning with CIFAR100 Krizhevsky et al. (2009) and Tiny-ImageNet Le & Yang (2015), we divide the dataset into ten different tasks, each containing data from $\{10, 20\}$ classes to simulate streaming tasks for two datasets; (2) Domain-Incremental Learning with Office31 Saenko et al. (2010) and Office-Caltech-10 Zhang & Davison (2020), we treat each domain as a separate task. Further dataset descriptions and preprocessing details are provided in Appendix A.1.

**Baselines.** To ensure a fair comparison with existing studies, we adopt the experimental protocols from Shenaj et al. (2023); Li et al. (2024a) for constructing AFCL tasks. We select baselines from three complementary perspectives. First, we include three asynchronous FL methods to benchmark the challenge of asynchronous aggregation: FedAsync Xie et al. (2019), FedBuff Nguyen et al. (2022b), and CA²FL Wang et al. (2024b). Second, we choose several representative FCL approaches as widely adopted baselines: GLFC Dong et al. (2022), FedCIL Qi et al. (2023), Re-Fed Li et al. (2024a), FOT Bakman et al. (2023), and FedSSI Li et al. (2025c). Finally, we consider the only existing AFCL method to highlight its contrast with our design: FedSpace Shenaj et al. (2023). Through comparisons across these baselines, we provide a comprehensive and rigorous validation of the effectiveness and superiority of our proposed method. Comprehensive descriptions of the baselines are provided in Appendix A.2.

**Configurations.** We configure each task with $E = 20$ local training epochs and $T = 100$ communication rounds, ensuring convergence before the introduction of the next task. The total number of clients is $K = 20$ with an active participation ratio of $r = 0.4$. We adopt ResNet18 He et al. (2016) as the backbone model. To introduce data heterogeneity, local samples are partitioned using a Dirichlet distribution $\text{Dir}(\alpha)$, where smaller $\alpha$ values correspond to higher Non-IID levels. For rehearsal-based FCL and asynchronous methods, each client is allocated a memory buffer of size 300 to store synthetic or previous samples for comparison fairness. To simulate the asynchronous setting, we randomly assign task arrival times to clients rather than enforcing global synchronization. Each participating client becomes available with a probability of $0.8$. We also allow update staleness of up to 25 rounds, which ensures persistent asynchrony while maintaining stable training. We evaluate performance by reporting the average accuracy $AC(\uparrow)$ and forgetting score $FS(\downarrow)$ over all tasks. Each experiment is repeated twice, and the average accuracy and standard deviation are computed from the last 10 rounds of each run. Optimization is performed using Adam with a linear learning rate schedule. All experiments are executed on a cluster with 24 RTX 4090 GPUs.

Table 1: Performance comparison of various methods on four datasets across two different incremental settings. We evaluate with two main metrics, and the best results are **bold**.

| Method | CIFAR100 | | Tiny-ImageNet | | Office31 | | Office-Caltech-10 | |
|---|---|---|---|---|---|---|---|---|
| | $AC(\uparrow)$ | $FS(\downarrow)$ | $AC(\uparrow)$ | $FS(\downarrow)$ | $AC(\uparrow)$ | $FS(\downarrow)$ | $AC(\uparrow)$ | $FS(\downarrow)$ |
| FedAsync | 30.66±2.32 | 37.91±5.15 | 28.64±3.37 | 45.78±2.78 | 41.97±1.07 | 31.45±5.19 | 43.68±3.16 | 28.25±4.30 |
| FedBuff | 29.50±3.43 | 42.01±1.04 | 26.15±3.17 | 44.36±5.64 | 39.61±2.59 | 39.81±4.41 | 40.42±2.93 | 35.24±3.22 |
| CA$^2$FL | 33.08±1.73 | 36.12±4.14 | 30.90±2.91 | 41.59±4.62 | 42.82±3.83 | 29.71±1.12 | 43.77±5.59 | 28.03±1.64 |
| GLFC | 31.11±2.35 | 33.56±1.97 | 27.71±4.75 | 35.20±1.99 | 44.03±3.47 | 22.65±1.86 | 45.92±3.01 | 22.01±1.89 |
| FedCIL | 31.93±3.56 | 38.22±1.17 | 31.52±4.61 | 41.63±2.40 | 45.96±3.59 | 25.11±3.82 | 47.41±0.64 | 25.43±1.69 |
| Re-Fed | 32.34±3.43 | 36.72±1.59 | 29.76±3.26 | 39.69±4.02 | 44.28±5.45 | 28.73±2.16 | 44.56±3.80 | 27.29±1.45 |
| FOT | 37.13±2.94 | 28.76±1.54 | 32.36±2.15 | 30.92±2.11 | 46.23±4.75 | 22.76±1.83 | 48.66±3.16 | 21.58±2.74 |
| FedSSI | 33.75±2.88 | 30.36±3.04 | 34.99±5.13 | 33.52±2.23 | 46.86±3.59 | 24.91±3.25 | 49.27±1.29 | 22.36±1.66 |
| FedSpace | 38.61±1.73 | 30.99±2.75 | 35.81±2.42 | 35.07±3.15 | 47.61±1.46 | 21.94±2.08 | 48.32±0.87 | 23.72±2.12 |
| **C$^2$-AFCL** | **42.57±2.93** | **27.36±1.31** | **38.24±2.90** | **30.11±1.95** | **50.54±3.16** | **20.79±2.06** | **52.66±2.44** | **20.32±3.71** |

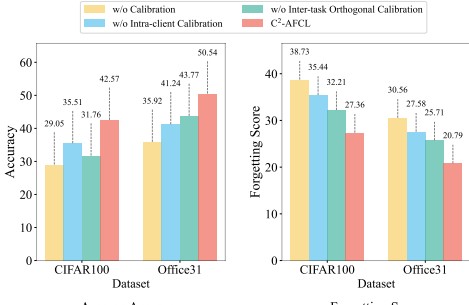

Figure 2: Ablation study of C$^2$-AFCL on two datasets with three components ($\alpha$=1).

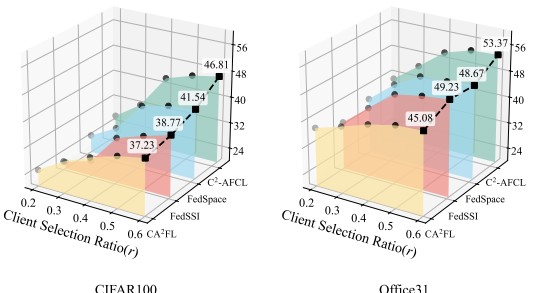

Figure 3: Performance comparison of various methods w.r.t. ratio $r$ between active clients and total clients in each round ($\alpha$=1).

## 6.2 PERFORMANCE OVERVIEW

**Main Results.** Table 1 shows that C$^2$-AFCL achieves the best balance between accuracy and forgetting across datasets and settings. Conventional asynchronous FL baselines like FedAsync fail to maintain stability under streaming tasks, while FCL methods reduce forgetting via replay or regularization but remain fragile in asynchronous environments due to their assumption of task alignment. FedSpace, the only existing AFCL method, improves stability through prototype-based calibration, yet its reliance on contrastive objectives limits its ability to decouple inter-task interference. In contrast, C$^2$-AFCL addresses both client and task drift while intra-client calibration aligns updates with true task progress, and inter-task orthogonal projection preserves historical knowledge. This mechanism enables continual learning with higher accuracy and lower forgetting across diverse scenarios.

**Ablation Study.** Figure 2 highlights the contributions of the two calibration modules in C$^2$-AFCL. Without intra-client calibration, updates are dominated by statistical and temporal variance, reducing stability. Without inter-task orthogonal calibration, new task gradients interfere with previous knowledge, leading to higher forgetting. The full framework updates capture genuine task progress while remaining orthogonal to historical knowledge, thereby maintaining a robust balance between stability and plasticity under asynchronous conditions.

**Communication Efficiency.** Table 2 indicates that C$^2$-AFCL attains favorable efficiency by converging within a comparable number of communication rounds while sustaining higher accuracy. This efficiency stems from the two-stage calibration: intra-client calibration filters out redundant local variance, reducing oscillations during aggregation, whereas inter-task calibration prevents destructive interference, ensuring that each update contributes meaningfully to the global model. As a result, the framework requires no additional communication overhead yet maintains stable optimization dynamics, achieving a better balance between convergence speed and final performance.

Table 2: We evaluate different methods based on the number of communication rounds required to reach their best test accuracy. Specifically, we report the total communication rounds needed to achieve the best performance on each task and analyze the trade-off between accuracy and communication cost. We further define "$\Delta$" as the difference between the percentage gain in accuracy and the percentage increase in communication rounds of $C^2$-AFCL compared to other baselines.

| Method | CIFAR100 | | Tiny-ImageNet | | Office31 | | Office-Caltech-10 | |
|---|---|---|---|---|---|---|---|---|
| | Rounds | $\Delta$ | Rounds | $\Delta$ | Rounds | $\Delta$ | Rounds | $\Delta$ |
| FedAsync | $841_{\pm2.17}$ | 37.07%↑ | $893_{\pm0.74}$ | 30.38%↑ | $213_{\pm1.53}$ | 20.89%↑ | $181_{\pm1.07}$ | 16.14%↑ |
| FedBuff | $820_{\pm0.94}$ | 39.91%↑ | $871_{\pm1.90}$ | 40.49%↑ | $199_{\pm1.45}$ | 21.06%↑ | $170_{\pm0.66}$ | 19.10%↑ |
| CA$^2$FL | $829_{\pm2.03}$ | 25.43%↑ | $909_{\pm1.44}$ | 22.43%↑ | $191_{\pm1.32}$ | 7.04%↑ | $177_{\pm1.71}$ | 13.53%↑ |
| GLFC | $849_{\pm2.12}$ | 36.02%↑ | $901_{\pm0.76}$ | 35.78%↑ | $219_{\pm1.83}$ | 17.99%↑ | $193_{\pm2.31}$ | 16.75%↑ |
| FedCIL | $810_{\pm2.47}$ | 27.64%↑ | $879_{\pm0.98}$ | 16.54%↑ | $204_{\pm0.94}$ | 6.05%↑ | $176_{\pm1.19}$ | 3.68%↑ |
| Re-Fed | $869_{\pm0.87}$ | 33.13%↑ | $896_{\pm2.93}$ | 25.70%↑ | $203_{\pm2.28}$ | 9.70%↑ | $192_{\pm0.49}$ | 19.74%↑ |
| FOT | $817_{\pm1.93}$ | 9.88%↑ | $865_{\pm2.24}$ | 11.70%↑ | $210_{\pm1.11}$ | 8.37%↑ | $180_{\pm1.30}$ | 3.22%↑ |
| FedSSI | $834_{\pm1.65}$ | 23.49%↑ | $924_{\pm1.09}$ | 9.61%↑ | $223_{\pm1.59}$ | 12.78%↑ | $185_{\pm0.91}$ | 4.72%↑ |
| FedSpace | $877_{\pm0.82}$ | 12.65%↑ | $946_{\pm0.74}$ | 9.43%↑ | $206_{\pm1.56}$ | 3.24%↑ | $197_{\pm1.60}$ | 13.04%↑ |
| $C^2$-AFCL | $856_{\pm1.69}$ | / | $921_{\pm1.92}$ | / | $212_{\pm1.43}$ | / | $189_{\pm2.01}$ | / |

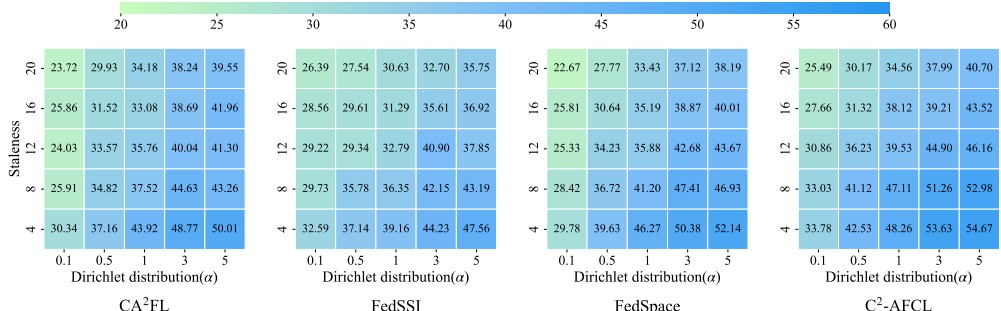

Figure 4: Performance comparison of various methods w.r.t. Dirichlet distribution $\mathrm{Dir}(\alpha)$ and staleness $\tau$ of asynchronous settings on CIFAR100.

**Parameter Sensitivity Analysis.** Figure 3 illustrates that $C^2$-AFCL remains robust across different client participation ratios. While other methods exhibit clear fluctuations when fewer clients are selected, our framework maintains stable accuracy. This robustness arises from the calibration design where intra-client calibration suppresses noise from sparse participation, and inter-task calibration ensures that limited updates still align with the preserved knowledge subspace. Thus, even under reduced client availability, the model preserves a stable learning trajectory without severe performance collapse. Figure 4 shows that $C^2$-AFCL adapts well to varying degrees of Non-IID distributions and asynchronous delays. Unlike baselines that accumulate bias as heterogeneity or staleness increases, our dual calibration effectively decomposes updates into informative and interfering components, filtering out the harmful directions. This prevents divergence caused by inconsistent task arrivals or skewed data partitions, leading to a smoother performance across challenging federated conditions.

## 7 CONCLUSION

In this work, we introduced $C^2$-AFCL, a cross-task calibration framework designed to address the fundamental challenge of asynchronous federated continual learning. Unlike prior works that assume synchronized task arrivals, our method explicitly handles both client drift and task drift through a dual orthogonal calibration strategy. We further provided a theoretical convergence analysis and demonstrated through extensive experiments that our approach consistently outperforms state-of-the-art baselines across diverse settings. We believe this work establishes AFCL as a practical and scalable paradigm for real-world federated continual learning and opens promising directions for future research in dynamic distributed environments.

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

## THE USE OF LARGE LANGUAGE MODELS (LLMS)

In preparing this work, we have used Large Language Models (LLMs) exclusively for the purposes of translation and language polishing. The content, arguments, and conclusions presented herein are entirely my own, and the use of LLMs did not contribute to the generation of original ideas or substantive content.

## ETHICS STATEMENT

This work complies with the ICLR Code of Ethics. No human subjects, personally identifiable data, or sensitive datasets were involved. The use of Large Language Models (LLMs) was strictly limited to translation and language polishing; they did not contribute to the generation of original ideas, methodology, results, or conclusions.

## REPRODUCIBILITY STATEMENT

We have taken steps to ensure the reproducibility of our work. The theoretical assumptions are explicitly stated in Section 5, with complete proofs provided in the Appendix. The experimental setup, including datasets, preprocessing steps, and hyperparameters, is detailed in Section 4 and the Appendix. Source code and instructions for reproducing our results are available if needed.

## A   EXPERIMENTAL SETTINGS

### A.1   DATASETS

**Class-Incremental Datasets**

**CIFAR-100** Krizhevsky et al. (2009) is a widely used benchmark for image classification, consisting of 60,000 images across 100 object categories with balanced class distributions. In our setting, we split the dataset into multiple class-incremental tasks, where each task contains 10 disjoint classes.

**Tiny-ImageNet** Le & Yang (2015) is a scaled-down version of ImageNet containing 200 classes with 500 training and 50 validation images per class at a resolution of $64 \times 64$. Following existing works, we partition the dataset into class-incremental tasks with 20 classes per task.

**Domain-Incremental Datasets**

**Office-31** Saenko et al. (2010) is a domain adaptation benchmark containing 4,652 images from 31 categories collected across three distinct domains: Amazon, Webcam, and DSLR. We adopt the domain-incremental setting by treating each domain as a separate task, enabling evaluation under distributional shifts.

**Office-Caltech-10** Zhang & Davison (2020) is a variant of the Office dataset that overlaps 10 common categories with the Caltech-256 dataset across four domains: Amazon, Webcam, DSLR, and Caltech. Each domain is regarded as a separate task in our experiments, providing a compact yet diverse benchmark for domain-incremental learning.

## A.2 BASELINES

**Representative Asynchronous Federated Learning**

**FedAsync** Xie et al. (2019) is an asynchronous variant of federated learning where the server immediately integrates each client update upon arrival. It mitigates the idle time caused by synchronization but suffers from update staleness when clients operate at different speeds. This method serves as a fundamental baseline for evaluating asynchronous aggregation schemes.

**FedBuff** Nguyen et al. (2022b) introduces a buffering mechanism that collects a fixed number of client updates before aggregation. By controlling the buffer size, it strikes a balance between staleness and synchronization efficiency. It is widely used as a strong asynchronous FL baseline in heterogeneous environments.

**CA$^2$FL** Wang et al. (2024b) enhances asynchronous FL by caching the latest client updates and calibrating incoming updates against them. This calibration reduces the impact of client and data heterogeneity on the aggregated global model. It provides a more stable alternative to purely staleness-based methods like FedAsync.

**Federated Continual Learning**

**GLFC** Dong et al. (2022) addresses federated class-incremental learning by maintaining a generative model to replay samples from previous tasks. This alleviates catastrophic forgetting without requiring raw data sharing among clients. It is one of the earliest works extending continual learning principles to the federated setting.

**FedCIL** Qi et al. (2023) improves federated class-incremental learning by integrating generative replay with class-balanced loss. This design tackles both catastrophic forgetting and class imbalance across clients. It is a representative baseline for continual learning with replay-based strategies.

**Re-Fed** Li et al. (2024a) leverages efficient replay mechanisms by selectively storing informative samples across clients. The method reduces memory usage while maintaining strong performance under federated continual learning. It highlights the trade-off between communication efficiency and forgetting mitigation.

**FOT** Bakman et al. (2023) prevents knowledge interference by projecting task updates onto orthogonal subspaces. This strategy preserves previously learned knowledge while integrating new tasks. It represents a regularization-based approach tailored for continual learning in federated environments.

**FedSSI** Li et al. (2025c) introduces a rehearsal-free continual learning method by extending synaptic intelligence to federated settings. It eliminates the need for data replay while retaining plasticity across sequential tasks. This makes it particularly suitable for privacy-sensitive or resource-constrained scenarios.

**Asynchronous Federated Continual Learning**

**FedSpace** Shenaj et al. (2023) is the first dedicated method for asynchronous federated continual learning. It employs fractal pre-training and prototype-based contrastive learning to alleviate task drift. This baseline directly targets the AFCL problem and provides a critical reference point for comparison with our method.

# B   Theoretical Analysis for C$^2$-AFCL

*Proof of Lemma 5.5.* Denote by $0 \leq \theta_1 \leq \cdots \leq \theta_d \leq \pi/2$ the principal angles between the subspaces $\text{Col}(b_d)$ and $\text{Col}(b_d^\top)$, and let $\sin \Theta = \text{diag}(\sin \theta_1, \ldots, \sin \theta_d)$. A standard formulation of the Davis–Kahan $\sin \Theta$ theorem gives $\|\sin \Theta\|_2 \leq \|E\|_2/\delta$.

The nonzero singular values of $\mathcal{P}^{(s+1)} - \mathcal{P}^{(s)}$ are exactly $\{\sin(2\theta_i)\}_{i=1}^d$. Hence

$$\|\mathcal{P}^{(s+1)} - \mathcal{P}^{(s)}\|_2 = \max_{1 \leq i \leq d} |\sin(2\theta_i)| = \sin(2\theta_{\max}),$$

where $\theta_{\max} = \max_i \theta_i$. Using $\sin(2x) = 2 \sin x \cos x \leq 2 \sin x$ for $x \in [0, \frac{\pi}{2}]$ we obtain

$$\|\mathcal{P}^{(s+1)} - \mathcal{P}^{(s)}\|_2 \leq 2 \sin \theta_{\max} = 2\|\sin \Theta\|_2 \leq \frac{2\|\mathcal{E}^{(s)}\|_2}{\delta}.$$

This proves the operator-norm bound.

Since $\mathcal{P}^{(s+1)} - \mathcal{P}^{(s)}$ is the difference of two rank-$d$ projectors, its rank is at most $2d$. For any matrix $M$ of rank $r$ we have $\|M\|_F \leq \sqrt{r}\, \|M\|_2$. Applying this with $M = \mathcal{P}^{(s+1)} - \mathcal{P}^{(s)}$ and $r \leq 2d$ yields

$$\|\mathcal{P}^{(s+1)} - \mathcal{P}^{(s)}\|_F \leq \sqrt{2d}\, \|\mathcal{P}^{(s+1)} - \mathcal{P}^{(s)}\|_2 \leq \sqrt{2d} \cdot \frac{2\|\mathcal{E}^{(s)}\|_2}{\delta} = \frac{2\sqrt{2d}\, \|\mathcal{E}^{(s)}\|_2}{\delta}.$$

Squaring both sides gives

$$\|\mathcal{P}^{(s+1)} - \mathcal{P}^{(s)}\|_F^2 \leq \frac{8d\, \|\mathcal{E}^{(s)}\|_2^2}{\delta^2},$$

which completes the proof.

*Proof of Theorem 5.6.* For the active task $t$ at round $r$, by $L$-smoothness,

$$F^t(w_{r+1}) \leq F^t(w_r) + \langle \nabla F^t(w_r),\, w_{r+1} - w_r \rangle + \frac{L}{2}\|w_{r+1} - w_r\|^2.$$

With $w_{r+1} - w_r = \eta g_r$ and $g_r = \frac{1}{K}\sum_{k=1}^K \delta_{\text{pre},k}^t$ this gives:

$$F^t(w_{r+1}) \leq F^t(w_r) + \eta\langle \nabla F^t(w_r), g_r \rangle + \frac{L\eta^2}{2}\|g_r\|^2. \tag{9}$$

Take full expectation and denote $a_r = (\mathbb{I} - \mathcal{P}_t)\nabla F^t(w_r)$ and $\bar{g}_r = \mathbb{E}[g_r \mid w_r]$. Because $g_r$ lies in the range of $(\mathbb{I} - \mathcal{P}_t)$ we have $\langle \nabla F^t(w_r), g_r \rangle = \langle a_r, g_r \rangle$. Hence,

$$\mathbb{E}[F^t(w_{r+1})] \leq \mathbb{E}[F^t(w_r)] + \eta\, \mathbb{E}[\langle a_r, g_r \rangle] + \frac{L\eta^2}{2}\mathbb{E}\|g_r\|^2. \tag{10}$$

Use $\langle u, v \rangle = \|u\|^2 + \langle u, v - u \rangle$ and the inequality $\langle u, v - u \rangle \geq -\frac{1}{2}\|u\|^2 - \frac{1}{2}\|v - u\|^2$. Taking expectation and using $\mathbb{E}[\langle a_r, g_r \rangle] = \mathbb{E}[\langle a_r, \bar{g}_r \rangle]$ we can get:

$$\mathbb{E}[\langle a_r, g_r \rangle] \geq \tfrac{1}{2}\mathbb{E}\|a_r\|^2 - \tfrac{1}{2}\mathbb{E}\|\bar{g}_r - a_r\|^2. \tag{11}$$

Substitute equation 11 into equation 10 and rearrange:

$$\eta\left(\tfrac{1}{2}\mathbb{E}\|a_r\|^2 - \tfrac{1}{2}\mathbb{E}\|\bar{g}_r - a_r\|^2\right) \leq \mathbb{E}[F^t(w_r)] - \mathbb{E}[F^t(w_{r+1})] + \frac{L\eta^2}{2}\mathbb{E}\|g_r\|^2.$$

Multiply by $2/\eta$ to obtain the central one-step inequality:

$$\mathbb{E}\|a_r\|^2 \leq \frac{2}{\eta}\Big(\mathbb{E}[F^t(w_r)] - \mathbb{E}[F^t(w_{r+1})]\Big) + \mathbb{E}\|\bar{g}_r - a_r\|^2 + L\eta\, \mathbb{E}\|g_r\|^2. \tag{12}$$

Let $\widetilde{g}_r = g_r - \bar{g}_r$. Then,

$$\|g_r\|^2 = \|\bar{g}_r + \widetilde{g}_r\|^2 = \|\bar{g}_r\|^2 + 2\langle \bar{g}_r, \widetilde{g}_r \rangle + \|\widetilde{g}_r\|^2.$$

Taking the conditional expectation given $w_r$,

$$\mathbb{E}\big[\|g_r\|^2 \mid w_r\big] = \|\bar{g}_r\|^2 + 2\,\mathbb{E}[\langle \bar{g}_r, \widetilde{g}_r\rangle \mid w_r] + \mathbb{E}\big[\|\widetilde{g}_r\|^2 \mid w_r\big].$$

Since $\bar{g}_r$ is $w_r$-measurable and $\mathbb{E}[\widetilde{g}_r \mid w_r] = 0$, the cross term vanishes:

$$\mathbb{E}[\langle \bar{g}_r, \widetilde{g}_r\rangle \mid w_r] = \big\langle \bar{g}_r, \mathbb{E}[\widetilde{g}_r \mid w_r]\big\rangle = \langle \bar{g}_r, 0\rangle = 0.$$

Taking expectation over $w_r$ gives:

$$\mathbb{E}\|g_r\|^2 = \mathbb{E}\|\bar{g}_r\|^2 + \mathbb{E}\|g_r - \bar{g}_r\|^2.$$

The average variance of client increments bounds the variance term. Since $\mathrm{Var}(\delta_{\mathrm{pre},k}^t) \le \mathrm{Var}(\delta_k^t) \le \sigma^2$ and clients are averaged:

$$\mathbb{E}\|g_r - \bar{g}_r\|^2 \le \frac{1}{K^2}\sum_{k=1}^{K}\sigma^2 = \frac{\sigma^2}{K}.$$

For $\|\bar{g}_r\|^2$ use $\bar{g}_r = a_r + (\bar{g}_r - a_r)$ and $(x+y)^2 \le 2x^2 + 2y^2$:

$$\|\bar{g}_r\|^2 \le 2\|a_r\|^2 + 2\|\bar{g}_r - a_r\|^2.$$

Thus,

$$\mathbb{E}\|g_r\|^2 \le \frac{\sigma^2}{K} + 2\mathbb{E}\|a_r\|^2 + 2\mathbb{E}\|\bar{g}_r - a_r\|^2. \tag{13}$$

Insert equation 13 into equation 12:

$$\mathbb{E}\|a_r\|^2 \le \frac{2}{\eta}\Delta_r + \mathbb{E}\|\bar{g}_r - a_r\|^2 + L\eta\Big(\frac{\sigma^2}{K} + 2\mathbb{E}\|a_r\|^2 + 2\mathbb{E}\|\bar{g}_r - a_r\|^2\Big).$$

where $\Delta_r = \mathbb{E}[F^t(w_r)] - \mathbb{E}[F^t(w_{r+1})]$. Rearrange left-hand side terms:

$$(1 - 2L\eta)\mathbb{E}\|a_r\|^2 \le \frac{2}{\eta}\Delta_r + (1 + 2L\eta)\mathbb{E}\|\bar{g}_r - a_r\|^2 + \frac{L\eta}{K}\sigma^2.$$

With the step-size condition $\eta \le 1/(4L)$ we have $2L\eta \le 1/2$ and the crude bounds:

$$\frac{1}{1 - 2L\eta} \le 2, \qquad \frac{1 + 2L\eta}{1 - 2L\eta} \le 3.$$

Dividing both sides by $1 - 2L\eta$ gives

$$\mathbb{E}\|a_r\|^2 \le \frac{4}{\eta}\Delta_r + 3\,\mathbb{E}\|\bar{g}_r - a_r\|^2 + \frac{2L\eta}{K}\sigma^2. \tag{14}$$

Denote $m_k^r = \mathbb{E}[\delta_k^t \mid w_r]$, using the operator-norm inequality and the fact that $(\mathbb{I} - \mathcal{P}_t)$ is an orthogonal projector, we can get:

$$\|\bar{g}_r - a_r\| = \|(\mathbb{I} - \mathcal{P}_t)\Big(\frac{1}{K}\sum_{k=1}^{K} m_k^r - \nabla F^t(w_r)\Big)\| \le \Big\|\frac{1}{K}\sum_{k=1}^{K} m_k^r - \nabla F^t(w^r)\Big\|.$$

Decompose

$$\frac{1}{K}\sum_{k=1}^{K} m_k^r - \nabla F^t(w_r) = \Big(\frac{1}{K}\sum_k (m_k^r - \nabla F_k^t(w_r))\Big) + \Big(\frac{1}{K}\sum_k \nabla F_k^t(w_r) - \nabla F^t(w_r)\Big).$$

Apply $(u+v)^2 \le 2u^2 + 2v^2$ and take expectation to get:

$$\mathbb{E}\|\bar{g}_r - a_r\|^2 \le 2\mathbb{E}\Big\|\frac{1}{K}\sum_{k=1}^{K}(m_k^r - \nabla F_k^t(w_r))\Big\|^2 + 2\mathbb{E}\Big\|\frac{1}{K}\sum_{k=1}^{K}\nabla F_k^t(w_r) - \nabla F^t(w_r)\Big\|^2.$$

By Assumption 5.2, the second term is at most $2\sigma_g^2$:

$$\mathbb{E}\Big\|\frac{1}{K}\sum_{k=1}^{K}\nabla F_k^t(w_r) - \nabla F^t(w_r)\Big\|^2 \le \sigma_g^2 \implies 2\sigma_g^2 \text{ in the bound.}$$

For the first term note each $m_k^r$ is computed at some delayed model $w_{r-\tau_k}$ with $\tau_k \le \tau_{\max}$. We now introduce the modeling error: $\varepsilon_k^r = m_k^r - \nabla F_k^t(w_{r-\tau_k})$ and then we can get:

$$m_k^r - \nabla F_k^t(w_r) = \big(m_k^r - \nabla F_k^t(w_{r-\tau_k})\big) + \big(\nabla F_k^t(w_{r-\tau_k}) - \nabla F_k^t(w_r)\big)$$
$$= \varepsilon_k^r + \big(\nabla F_k^t(w_{r-\tau_k}) - \nabla F_k^t(w_r)\big).$$

By Assumption 5.1, the gradient difference is bounded as:

$$\|m_k^r - \nabla F_k^t(w_r)\| \le L\|w_{r-\tau_k} - w_r\| + \|\varepsilon_k^r\|.$$

We assume the client's expected increment equals the stale-point gradient, and $w_r - w_{r-\tau_k} = \sum_{j=r-\tau_k}^{r-1}\eta g_j$, so $\|m_k^r - \nabla F_k^t(w_r)\| \le L\eta\sum_{j=r-\tau_k}^{r-1}\|g_j\|$. Using $\tau_k \le \tau_{\max}$ and Cauchy–Schwarz we get the squared bound:

$$\|m_k^r - \nabla F_k^t(w_r)\|^2 \le L^2\eta^2\tau_{\max}\sum_{j=r-\tau_k}^{r-1}\|g_j\|^2.$$

Taking expectations and averaging over $k$ iterations and the standard conservative bound:

$$\mathbb{E}\Big\|\frac{1}{K}\sum_{k=1}^{K}(m_k^r - \nabla F_k^t(w_r))\Big\|^2 \le L^2\eta^2\tau_{\max}^2\Big(G^2 + \frac{\sigma^2}{K}\Big).$$

Thus, the bias bound becomes:

$$\mathbb{E}\|\bar{g}_r - a_r\|^2 \le 2L^2\eta^2\tau_{\max}^2\Big(G^2 + \frac{\sigma^2}{K}\Big) + 2\sigma_g^2. \tag{15}$$

Put equation 15 into equation 14. Substitute to get:

$$\mathbb{E}\|a^r\|^2 \le \frac{4}{\eta}\Delta_r + 6L^2\eta^2\tau_{\max}^2 G^2 + 6\sigma_g^2 + \frac{2L\eta\sigma^2(3L\eta\tau_{max}^2 + 1)}{K}. \tag{16}$$

Summing equation 16 over $r = 0, \dots, R-1$ and dividing by $R$ iterations:

$$\frac{1}{R}\sum_{r=0}^{R-1}\mathbb{E}\|a_r\|^2 \le \frac{4}{\eta R}\sum_{r=0}^{R-1}\Delta_r + 6L^2\eta^2\tau_{\max}^2 G^2 + 6\sigma_g^2 + \frac{2L\eta\sigma^2(3L\eta\tau_{max}^2 + 1)}{K}.$$

Using $\sum_{r=0}^{R-1}\Delta_r = \mathbb{E}[F^t(w_0)] - \mathbb{E}[F^t(w_R)] \le F^t(w_0) - F^t(w^\star)$, we can obtain:

$$\frac{1}{R}\sum_{r=0}^{R-1}\mathbb{E}\|a_r\|^2 \le \frac{4\big(F^t(w_0) - F^t(w^\star)\big)}{\eta R} + 6L^2\eta^2\tau_{\max}^2 G^2 + 6\sigma_g^2 + \frac{2L\eta\sigma^2(3L\eta\tau_{max}^2 + 1)}{K}.$$

The above derivation assumed the projector $\mathcal{P}_t$ used in definitions of $a_r$ and in $(\mathbb{I} - \mathcal{P}_t)\delta_k^t$ is fixed. When $\mathcal{P}_t$ is updated periodically, there is a mismatch between the projector used to produce some historical $\delta_k^t$ and the projector used in the inner product; this mismatch yields an additional additive term which can be shown (see Lemma 5.5 below) to be upper-bounded by $3G^2\Delta_{\text{proj}}$ in the average. Adding this term to the right-hand side yields the claimed bound:

$$\frac{1}{R}\sum_{r=0}^{R-1}\mathbb{E}\big\|(\mathbb{I} - \mathcal{P}_t)\nabla F^t(w_r)\big\|^2 \le \frac{4\big(F^t(w_0) - F^t(w^\star)\big)}{\eta R} + 6L^2\eta^2\tau_{\max}^2 G^2 + 6\sigma_g^2$$
$$+ \frac{2L\eta\sigma^2(3L\eta\tau_{max}^2 + 1)}{K} + 3G^2\Delta_{\text{proj}}^t.$$

This completes the proof.