# OpenReview forum: "C$^2$-AFCL: Cross-task Calibration for Asynchronous Federated Continual Learning"
_ICLR.cc/2026/Conference — Submitted to ICLR 2026_

### Official Review · Reviewer_taPG · 2025-10-26

**Soundness:** 2
**Presentation:** 1
**Contribution:** 2
**Rating:** 0
**Confidence:** 5

**Summary:**

The paper provides a method to decompose the gradients into two components, which are preserving and interference components. By reducing the interference components and update towards the preserving components only, the system will optimize towards the good direction only.

**Strengths:**

Although the authors proposed and considered the good research questions in FCL, the idea of decomposition the gradients into two subspace is potential. There are still issues remaining. Please check the weaknesses for more details.

**Weaknesses:**

1. The writing requires significant improvement. For instance,
   - There are a lot of inconsistent writings (e.g., the Federated Continual Learning and FCL; continual learning and CL being used arbitrarily).
   - Many statements are not clear/correct and not being well-supported with the references. For example,
        - L84-L85: while the asynchronous FCL is not clearly discussed (with only 1 paper discussed), the authors referred to a survey and mention that the direction is a potential future work, which making the readers unclear about the concept, also, redundant information making readers confused.
        - The works of "asynchronous FCL" is actually in CVPRW, not CVPR. This makes the authors' statements not strong enough to show that the research direction is important in current literature, as the worked is not peer-reviewed from main venue.
        - In L140, authors said that continual learning is also known as incremental learning, which is not totally correct.
    - The notations are inconsistent and not well-writing. For instance,
        - The writing $x\sim P(x|y)\in D^j$ in L179 seems to be verbose.
1. The terminology of client-drift and task-drift should be carefully discussed with good references to support. Furthermore, the task-drift and client-drift seem to be overlap in terms of meaning. Also, as the authors considered the different tasks from different clients, the works should be consider the heterogeneous FCL along with asynchronous FCL.
1. The literature reviews and baselines are not up-to-date. Many SOTA are missing.
1. The works seem to store a lot of information in both local memory and global memory, which makes the work lack novelty and seems to be impractical in real-work scenarios.
1. The bounds in L328 seem to be loose, making the Theorem 5.6 not significant. Furthermore, the contribution of the theorem proof is limited. To be more specific, except denote $a_r = \nabla F^t(w_r)$ in L792, without any careful discussion, the other parts of proof can easily be found in other works of FL. Furthermore, it is unclear how $g_r$ lies in the range of $\mathbb{I} - \mathcal{P}_t$. Furthermore, although the authors provided the convergence of vanilla FL, it is unclear to see the characteristics of FCL in the Theorem.
1. In L426, communication efficiency does not seem to be well-represented by the number of communication rounds. Please note that the communication cost can be represented as communication rounds $\times$ cost per round.
1. Many references are not peer-reviewed, up-to-date or updated the earliest versions, making the papers' statements not well-supported.

**Questions:**

1. In Figure 1, the authors seem to consider the combination between heterogeneous FCL and asynchronous FCL. Is it correct?
2. Please discuss carefully about the aggregation in Eq. (2). This is somehow uncommon with conventional FL.
2. In Eq. (3), we only regularize between task $t$ and $\tau$?
2. In L204, does $\tau$ represent all tasks?
2. In L209, "task heterogeneity challenges" inconsistent with the challenges shown in the introduction.
2. Can you explain carefully about the calibrated update in L8-Algo1? Why from this update, we can update the model as in L14-Algo1? Also, in L255-L260, the authors claimed that the $\delta$ is a more accurate representation of clients' immediate learning trajectory? How can be it? Please note that $\delta$ is the substraction of two gradient of same tasks, it refers more to be the variance. Furthermore, how can we reduce variance as mentioned in L260?
2. In L11-Algo1, how can we achieve $b_d$ from top-d basis? Furthermore, as $\mathcal{P}_t$ is the projection matrix, so is $b_d$ is orthogonal? How are the features orthogonal? Do you mean the $\mathcal{B}_d$, not $b_d$?
2. What are the rationales behind Eq. (5)? How can we define the subspace given by $P_t$ is the interference component, and harmful to previous tasks? Why the $\mathbb{I} - P_t$ is the novel knowledge?
2. In L273, the authors used $a_t$. However, this direction is very sensitive to the time-step of the learning model. Please explain? Furthermore, $N$ update direction is not carefully discussed.
2. Does the server also store the informations $\mathcal{A}$? Also, in L253, does the server also cach the task-aware update for all clients and all tasks in clients? This has the size of $K \times T \times D + T\times D$, where $D$ is the model size, and have a very large amount of memory.
2. The paper only compare methods with ResNet18, which is small and can not show the generalization of the proposed method.
2. What are the settings for the Table 1?

---

### Official Review · Reviewer_cMF5 · 2025-10-28

**Soundness:** 2
**Presentation:** 1
**Contribution:** 2
**Rating:** 2
**Confidence:** 3

**Summary:**

The paper introduces a cross-task calibration technique in the asynchronous FL setting in 2 stages. The first stage addresses intra-client task drift by using task-aware caches. The second stage addresses inter-client interference using techniques inspired by the gradient projection memory approach but adapted to asynchronous FL.

**Strengths:**

The paper addresses a less attended area, i.e., asynchronous federated learning and address the task and data drifts within and across clients.

The extension of GPM approach to federated (and asynchronous updates) setting has merits as GPM like techniques have shown success in ensuring new gradient updates are not interfering with performance on past tasks.

**Weaknesses:**

My main concern is that the paper borrows heavily from the “Gradient Projection Memory for Continual learning, ICLR 2021”, especially for the inter-client calibration. However, there is no mention of this and the paper reads as if the idea was first time proposed here.

Notations in Equation 1 seem inconsistent. Specifically as it minimizes over w^t while the loss depends on w_k^j from past tasks, which are independent variables—making the objective ill-defined. Additionally, the expectation term is misused: the former should be over (x,y)\!\sim\!D_k^j. The indices for time and task seem to be overlapping.

The paper uses mean deltas from weights in order to estimate the subspace. The original approach of GPM uses covariance-based subspace and on a per-layer level. The paper needs to provide clear reasoning for why the mean deltas approach works better or why is it practical?

**Questions:**

The paper uses mean deltas from weights in order to estimate the subspace. The original approach of GPM uses covariance-based subspace and on a per-layer level. The paper needs to provide clear reasoning for why the mean deltas approach works better or why is it practical?

I am a little confused about the intra client calibration. Why does subtracting state updates make sense? What if the global model has drifted significantly since the last cached update?

---

### Official Review · Reviewer_nSyN · 2025-10-30

**Soundness:** 3
**Presentation:** 3
**Contribution:** 2
**Rating:** 6
**Confidence:** 5

**Summary:**

This paper addresses the challenge of asynchronous federated continual learning (AFCL) and introduces the C$^2$-AFCL framework. The framework employs a two-stage orthogonal calibration mechanism that advances update calibration beyond statistical variance reduction toward semantic-level knowledge, making it the first to address task drift at a semantic level within an AFCL setting.

**Strengths:**

1. The paper is clearly organized and highly readable. Its motivation is a meaningful research direction in federated learning, and, for the first time, it addresses task drift in AFCL at the semantic level.

2. The convergence proof is solid, supported by a well-developed theoretical analysis.

3. Comprehensive experiments on diverse datasets and heterogeneity levels, complemented by detailed ablation analyses and thorough evaluations of communication efficiency.

**Weaknesses:**

1. Regarding scalability, the interference subspace is constructed from all past tasks, and performing SVD on this ever-growing set of vectors may become a bottleneck in long-term learning scenarios, warranting further investigation.

2. The interference subspace is obtained from the average update vectors of all past tasks. Treating all vectors indiscriminately in this way may provide insufficient support for semantic-level analysis.

3. This  contains several writing errors, such as xx.xx% (line 108).

4. The paper's problem definition is ambiguous, failing to provide a rigorous or formal mathematical definition of Asynchronous Federated Continuous Learning (AFCL). Its setup, assumptions, and distinctions from traditional FCL or asynchronous FL are not clearly articulated.

5. Some sections (such as problem statements and theoretical analysis) lack clear transitions and detailed explanations. Symbol usage occasionally shows inconsistencies, slightly affecting readability.

Overall, the quality of this paper is good, and I hope the authors can address my questions.

**Questions:**

1. In the experimental setup, the maximum allowed staleness is 25 rounds, while each task involves 100 communication rounds, which is relatively small. Would the method still remain effective if staleness were further increased?

---

### Meta-Review · Area_Chair_gfWh · 2026-01-04

**Summary:**

The paper received a strong reject, a reject, and a weak accept (only three reviewers submitted their reviews). The authors **did not submit a rebuttal**. As a result, this is a straightforward reject case, but I will summarize the main arguments given there was a weak accept opinion.

1) A reviewer (cMF5) indicated that the paper borrows heavily from the “Gradient Projection Memory for Continual learning, ICLR 2021”, however, there is no mention of this and the paper reads as if the idea was first time proposed here. The difference from that original paper is not clearly motivated also: the same reviewer points out that the paper uses mean deltas from weights in order to estimate the subspace, while the original approach of GPM uses covariance-based subspace and on a per-layer level. It is not clear why the former approach is better (is it, e.g., more efficient?)

2) Presentation issues regarding to assumptions stated, inconsistent notation, claims rigorously supported,  standard terminology being misused, etc. were brought up by all three reviewers, including the positive one, even though the latter claimed that "(o)verall, the quality of this paper is good".

3) The bounds of the theorem were deemed too loose by the most negative reviewer (taPG).

**Reviewer Concerns:**

None of the concerns raised were addressed by the rebuttal, as there was no rebuttal.

**Reviewer Scores:**

Reviewers would not have changed their score.

---

### Decision · Program_Chairs · 2026-01-26

Reject